# Exploring the Anti-Osteoporotic Potential of Daucosterol: Impact on Osteoclast and Osteoblast Activities

**DOI:** 10.3390/ijms242216465

**Published:** 2023-11-17

**Authors:** Sumin Lee, Jae-Hyun Kim, Minsun Kim, Sooyeon Hong, Hoyeon Park, Eom Ji Kim, Eun-Young Kim, Chungho Lee, Youngjoo Sohn, Hyuk Sang Jung

**Affiliations:** Department of Anatomy, College of Korean Medicine, Kyung Hee University, 26, Kyunghee dae-ro, Dongdaemun-gu, Seoul 02-447, Republic of Korea; 00sumin0209@naver.com (S.L.); jhk1@khu.ac.kr (J.-H.K.); alstjs8644@naver.com (M.K.); ghdtndus121@naver.com (S.H.); gmaad1215@naver.com (H.P.); dja1624513@naver.com (E.J.K.); turns@hanmail.net (E.-Y.K.); heavenmedi@naver.com (C.L.); youngjoos@khu.ac.kr (Y.S.)

**Keywords:** daucosterol, bone metabolism, osteoporosis, osteoclast, osteoblast

## Abstract

Osteoporosis is a debilitating condition characterized by reduced bone mass and density, leading to compromised structural integrity of the bones. While conventional treatments, such as bisphosphonates and selective estrogen receptor modulators (SERMs), have been employed to mitigate bone loss, their effectiveness is often compromised by a spectrum of adverse side effects, ranging from gastrointestinal discomfort and musculoskeletal pain to more severe concerns like atypical fractures and hormonal imbalances. Daucosterol (DC), a natural compound derived from various plant sources, has recently garnered considerable attention in the field of pharmacology. In this study, we investigated the anti-osteoporosis potential of DC by characterizing its role in osteoclasts, osteoblasts, and lipopolysaccharide (LPS)-induced osteoporosis. The inhibitory effect of DC on osteoclast differentiation was determined by tartrate-resistant acid phosphatase (TRAP) staining, F-actin ring formation by fluorescent staining, and bone resorption by pit formation assay. In addition, the calcification nodule deposition effect of osteoblasts was determined by Alizarin red S staining. The effective mechanisms of both cells were verified by Western blot and reverse transcription polymerase chain reaction (RT-PCR). To confirm the effect of DC in vivo, DC was administered to a model of osteoporosis by intraperitoneal administration of LPS. The anti-osteoporosis effect was then characterized by micro-CT and serum analysis. The results showed that DC effectively inhibited osteoclast differentiation at an early stage, promoted osteoblast activity, and inhibited LPS-induced bone density loss. The results of this study suggest that DC can treat osteoporosis through osteoclast and osteoblast regulation, and therefore may be considered as a new therapeutic alternative for osteoporosis patients in the future.

## 1. Introduction

Osteoporosis is a debilitating condition characterized by reduced bone mass and density, leading to the compromised structural integrity of the bones [1]. It primarily affects the elderly population and significantly increases the risk of fractures, especially in the spine and hip regions, leading to a marked decline in the quality of life for affected individuals [2]. Maintaining a healthy balance in bone metabolism, regulated by the intricate interplay between osteoclasts and osteoblasts, is paramount for preventing the onset and progression of osteoporosis [3]. While conventional treatments, such as bisphosphonates and selective estrogen receptor modulators (SERMs), have been employed to mitigate bone loss, their effectiveness is often compromised by a spectrum of adverse side effects, ranging from gastrointestinal discomfort and musculoskeletal pain to more severe concerns like atypical fractures and hormonal imbalances [4]. Furthermore, their long-term efficacy remains limited, with diminishing returns over extended treatment periods, necessitating a critical reevaluation of their role in osteoporosis management [4]. Thus, there is a pressing need for the development of novel therapeutic approaches to address these challenges and provide better management options for osteoporosis.

Osteoclast differentiation is initiated by the binding of receptor activator of nuclear factor-kappa B ligand (RANKL) to its receptor activator of nuclear factor-kappa B (RANK), followed by the activation of mitogen-activated protein kinase (MAPK) signaling and subsequent induction of c-Fos expression [5]. c-Fos then collaborates with other transcription factors to stimulate the nuclear factor of activated T cell c1 (NFATc1) expression, ultimately promoting osteoclast formation [6]. In contrast, osteoblast differentiation begins with bone morphogenetic protein-2 (BMP-2) signaling, which activates Smad proteins [7]. Activated Smads translocate to the nucleus and interact with runt-related transcription factor 2 (RUNX2) [7]. Additionally, wingless-related integration site (Wnt)/β-catenin is also a key factor in osteoblast differentiation, upregulating RUNX2 expression and contributing to mature osteoblast formation [8]. Understanding the intricacies of these differentiation mechanisms is crucial for developing therapeutic interventions aimed at targeting bone diseases.

Daucosterol (DC), a natural compound derived from various plant sources, has recently garnered considerable attention in the field of pharmacology [9]. This steroidal glycoside exhibits a wide range of biological activities and has shown promise in addressing various health-related challenges. Over the years, researchers have identified DC as a bioactive compound with diverse biological properties, including anti-cancer [10], anti-hepatic injury [11], antioxidant [12], and immunomodulatory effects [13]. Various recent studies have shown that natural compounds with anti-inflammatory effects may be effective against osteoclasts and osteoporosis through inhibition of the expression of RANKL and inflammatory cytokines [14,15]. In addition, the inhibition of reactive oxygen species (ROS) through antioxidant action inhibits the activity of osteoclasts and activates osteoblasts [16]. For these reasons, the various pharmacological effects demonstrated in previous studies of DC were expected to make them suitable as an effective drug for bone metabolism. However, the effectiveness of DC in osteoclasts and osteoblasts has not yet been verified. This research aims to delve deeper into the pharmacological actions of DC and elucidate its suitability as a therapeutic agent for bone diseases, thereby contributing to the advancement of musculoskeletal medicine.

## 2. Results

### 2.1. DC Inhibits Osteoclast Differentiation and Expression of the Specific Marker Tartrate-Resistant Acid Phosphatase (TRAP)

Before investigating the pharmacological effects of DC on the inhibition of osteoclast differentiation, we tested the cytotoxicity of DC on RAW 264.7 cells and osteoclasts. Consistent with the experiments, DC did not exhibit cytotoxicity in RAW 264.7 cells (Figure 1A), nor did they exhibit cytotoxicity in RANKL-induced osteoclasts (Figure 1B). Inhibition of osteoclast differentiation is considered a major therapeutic target for inhibiting and treating the exacerbation of osteoporosis. TRAP is an osteoclast-specific marker that shows a reddish-positive reaction to multinucleated red giant cells when stained [17]. In Figure 1C, RANKL-treated cells differentiated into red, giant multinucleated cells. DC inhibited this differentiation, especially 200 μg/mL of DC, which resulted in a cell size and color similar to RANKL (−). Counting the number of differentiated cells, RANKL induced a significant increase in the number of osteoclasts, and DC inhibited this increase in a concentration-dependent manner (Figure 1D). Furthermore, measuring the activity of TRAP in the culture medium, DC significantly inhibited the activity of TRAP induced by RANKL in a concentration-dependent manner, consistent with the staining results (Figure 1E). Furthermore, the effect of DC on the expression of TRAP mRNA (gene name: *Acp5*) was verified (Figure 1F). The expression of TRAP was quantified by β-actin (gene name: *Actb*), and DC inhibited the mRNA expression of TRAP in a concentration-dependent manner (Figure 1G).

### 2.2. DC Suppresses the Expression of Osteoclast Core Transcription Factors c-Fos and NFATc1 through Inhibition of MAPK Phosphorylation

Osteoclasts are formed by the differentiation, fusion, and maturation of precursor cells. As each stage is induced by the expression of different factors, identifying the stage at which a drug is effective is important for understanding its osteoclast inhibitory effects. We treated DC for 0–2 days, 2–4 days, and 4–6 days to determine which time period was most effective. DC reduced the size and number of osteoclasts and attenuated their red color in all time periods treated, with the strongest inhibitory effect on osteoclast differentiation at the beginning of differentiation (days 0–2) (Figure 2A). Consistent with the staining results, TRAP activity in the culture medium also showed that DC were most effective at the early stage of differentiation (Figure 2B). Phosphorylation of MAPKs and nuclear translocation of NF-κB are induced immediately after RANKL stimulation [5]. This induces the key osteoclast transcription factors NFATc1 and c-Fos [6]. In Figure 2C, RANKL induces the phosphorylation of MAPKs and nuclear translocation of NF-κB, and DC inhibits these increases. When expression was quantified as total form of each factor and Lamin B, DC reduced expression in a concentration-dependent manner, with significant inhibition of all factors at 200 μg/mL (Figure 2D). To investigate the effect of DC on the protein and mRNA expression of NFATc1 (gene name: *Nfatc1*) and c-Fos (gene name: *Fos*), Western blot and RT-PCR were performed (Figure 2E,F). Each factor was strongly induced by stimulation of RANKL. The protein expression of NFATc1 and c-Fos was significantly inhibited by DC treatment at all concentrations, and the mRNA expression was also significantly inhibited, as the expression of each indicator was quantified by β-actin (*Actb*) (Figure 2G,H).

### 2.3. DC Inhibits F-Actin Ring Formation and Expression of Osteoclast Fusion Markers

The F-actin ring is a cytoskeletal structure that plays a major role in the movement of osteoclasts and the maintenance of the bone resorption environment. As shown in Figure 3A, RANKL induced the formation of F-actin rings, and DC inhibited their formation. When counted, DC significantly inhibited F-actin ring formation at all concentrations (Figure 3B). ATPase H+ Transporting V0 Subunit D2 (ATP6v0d2, gene name: *Atp6v0d2*) and dendritic cell-specific transmembrane protein (DC-STAMP, gene name: *Dcstamp*) are genes that play a major role in F-actin ring formation [18]. Their mRNA expression was increased by RANKL, and DC decreased this expression (Figure 3C). Expression was quantified by *Actb*, and 200 μg/mL of DC inhibited this expression (Figure 3D,E).

### 2.4. DC Reduces Pit Formation Formation and Expression of Bone Resorption Factors

Pit formation inhibition experiments are used in a variety of studies to determine the efficacy of osteoclast bone resorption inhibition under conditions that mimic in vivo bone metabolism. RANKL-induced induction of osteoclasts identifies pits (black arrows) that are not observed in RAW 264.7 cells, and DC inhibited the formation of these pits (Figure 4A). The area of these pits was measured using ImageJ, and DC showed a significant inhibitory effect at a concentration of 200 μg/mL compared to RANKL-treated cells (Figure 4B). The formation of pits is associated with the expression of bone resorption-related genes such as matrix metalloproteinase-9 (MMP-9, gene name: *Mmp9*), cathepsin K (CTsK, gene name: *Ctsk*), and carbonic anhydrase II (CA2, gene name: *Ca2*) [19]. To verify the effect of DC on the expression of these genes, RT-PCR was performed (Figure 4C). The expression of each gene was quantified by *Actb*, and *Mmp9*, *Ctsk*, and *Ca2* were upregulated by RANKL, and DC significantly inhibited the expression of *Mmp9* at 200 μg/mL (Figure 4D) and *Ctsk* at 100 and 200 μg/mL (Figure 4E). DC significantly reduced the expression of *Ca2* at 200 μg/mL (Figure 4F).

### 2.5. DC Promotes Osteoblast Differentiation and Calcific Nodule Formation through Upregulation of BMP-2/Smad and Wnt/β-Catenin Mechanisms

Promoting osteoblast activity is crucial to prevent worsening of advanced osteoporosis and to restore lost bone density. To determine the effect of DC on the toxicity of MC3T3-E1 cells, they were incubated for 24 h. The results showed that DC did not significantly affect the toxicity of MC3T3-E1 cells (Figure 5A). MC3T3-E1 cells were treated with medium containing ascorbic acid and β-glycerophosphate (differentiation medium; D.M) to induce osteoblast differentiation and calcified nodule formation. The reaction was then terminated at 14 and 17 days and the deposited calcification nodules were stained with Alizarin red S dye (Figure 5B). At 14 days of the reaction, DC 200 μg/mL exhibited a more rapid calcification nodule formation effect (Figure 5C), and at 17 days, a concentration-dependent increase in calcification nodule formation capacity was observed compared to D.M-treated cells. Osteoblast activity is primarily driven by the BMP-2/Smad and Wnt/β-catenin mechanisms [7,8]. The effect of DC on BMP-2/Smad mechanism-related protein expression was validated by Western blot (Figure 5D). The results showed that 100 and 200 μg/mL DC significantly enhanced all factors compared to untreated cells (Figure 5E). The effect of DC on Wnt/β-catenin mechanism-related genes was determined by RT-PCR (Figure 5F). The results showed that 200 μg/mL of DC significantly increased the mRNA expression of Wnt-related factors (Figure 5G). Furthermore, the effect of DC on osteoblast-related genes (alkaline phosphatase, ALP; osteocalcin, OCN; pro-alpha1 chains of type I collagen; procol1) was determined by RT-PCR (Figure 5H). It was confirmed that all factors were significantly increased at 200 μg/mL through DC treatment (Figure 5I).

### 2.6. DC Significantly Prevents LPS-Induced Bone Density Reduction and Microarchitectural Deterioration

Administration of LPS induces an increase in inflammatory cytokines in the body, which promotes osteoclast activity and induces osteoblast apoptosis, leading to osteoporosis [20]. In Figure 6A, LPS administration decreased bone mass in the femur. On the other hand, DC inhibited this decrease. Bone microstructure was analyzed using Micro-CT analysis software (SkyScan version 1.6.10.1), and bone volume/total volume (BV/TV) decreased after LPS administration; this decrease was inhibited by DC administration (Figure 6B). In addition, trabecular thickness (Tb.Th), trabecular number (Tb.N), and trabecular separation (Tb.Sp), which are indicators of bone microstructure, were deteriorated by LPS administration, and DC inhibited this deterioration (Figure 6C–F). In particular, the difference in Tb.N was significant. Structure model index (SMI) is a value that measures the relative number of rod-shaped and plate-shaped trabecular bone structures and is an indicator of bone strength. In this study, LPS increased SMI levels, and DC decreased this increase.

## 3. Discussion

Globally, according to an analysis by the McKinsey Health Institute (MHI), by 2050, the number of people aged 65 and older worldwide will more than double to 1.6 billion, or 16.5% of the total. As the global population ages, the management of chronic diseases such as osteoporosis becomes increasingly important. The occurrence of fractures due to osteoporosis severely impairs the quality of life of individuals and families, with a mortality rate of up to 30%, especially for femur fractures [1]. Appropriate healthcare during the osteopenia phase, before the onset of osteoporosis, is therefore suggested as a good way to mitigate future healthcare costs and system burden. However, a clear limitation of current osteoporosis treatments is the issue of side effects, with approximately 8% of patients recently being hospitalized for adverse reactions to synthetic drugs, and thousands of people losing their lives each year to over-the-counter drug-related deaths [21]. Statistics show that over 80% of both healthcare professionals and the general public prefer natural products, and as a result, interest in natural products is growing year on year due to the widespread belief that they have fewer side effects than synthetic drugs [22,23]. DC is a plant sterol and one of the main components found in plants. The role of DC in bone metabolism demonstrated in this study may open up a number of opportunities for the treatment of osteoporosis in the future. In this study, we identified five roles for DC in the regulation of bone metabolism. (i) DC inhibits osteoclast differentiation through NFATc1/c-Fos inhibition. (ii) DC is effective early in osteoclast differentiation, which is locked into a MAPKs/nuclear factor kappa-light-chain-enhancer of activated B cells (NF-κB) mechanism. (iii) DC inhibits F-actin formation and bone resorption capacity through the control of expression of osteoclast-related genes. (iv) DC promotes osteoblast differentiation and calcified nodule formation through BMP-2 and Wnt mechanisms. (v) DC inhibits LPS-induced osteolysis.

Osteoclast precursor cells go through the stages of differentiation, fusion, and maturation to become osteoclasts [5]. Each stage is characterized by the successive expression of different transcription factors, each of which plays a unique role. We sought to validate DC at different stages of differentiation. TRAP, an osteoclast-specific marker, is abundant in numerous cytoplasmic granules and promotes bone resorption by digesting extracellular matrix and dissolving mineral crystals [17]. In this study, DC suppressed the expression of TRAP, with particularly strong inhibition at the beginning of differentiation (day 0–2). Based on these results, we validated the phosphorylation of MAPKs and the nuclear translocation of NF-κB, a signal transduction system that is immediately expressed after the binding of RANKL to RANK [5]. Extracellular signal-regulated kinase (ERK) and p38 are serine/threonine kinases that transduce signals from extracellular stimuli to multiple substrates involved in cell growth, differentiation, and apoptosis [5]. In detail, ERK is required for the survival of hematopoietic cells and is a member of the MAPK family, which is important for cell growth, differentiation, and apoptosis [24]. Among the osteogenic hormones, macrophage colony-stimulating factor (M-CSF), interleukin-1 (IL-1), and tumor necrosis factor-α (TNF-α) induce ERK phosphorylation, which prolongs osteoclast survival. p38 has also been shown to be involved in osteoclast bone resorption induced by IL-1 and TNF-α in the fetal long bone. However, some recent studies have suggested that p38 plays a major role in osteoclast differentiation, but not in bone resorption [25]. NF-κB is a versatile transcription factor that plays a role in regulating osteoclast development, activity, and longevity [26]. The canonical NF-κB pathway is typically activated by ligand binding to receptors on the cell surface such as RANK, TNFR, and IL-1R, which promotes nuclear translocation of NF-κB through degradation of IκB [26]. Subsequently, it induces the activity of various osteoclast differentiation essential transcription factors. In the present study, DC tended to suppress the expression of all three markers. Similar to the TRAP results, this indicates that DC played an effective role in the early stages of osteoclast differentiation, and that a delay in differentiation itself may lead to impairment in the fusion and maturation stages. Expression of both mechanisms sequentially induces c-Fos and NFATc1, which are known to be key transcription factors for osteoclast differentiation [6]. c-Fos is a major player in the activator protein 1 (AP-1) heterodimer and is expressed by RANKL, and c-Fos knock-out cells are deficient in osteoclastogenesis [27]. It was also observed that embryonic stem cells lacking NFATc1 were incapable of osteoclastogenic activity. In addition, overexpression of NFATc1 in RANKL-deficient cells induced osteoclasts [28]. Both factors control a variety of osteoclast-related factors. NFATc1 controls the expression of osteoclast fusion-related factors ATP6v0d2 and DC-STAMP [29], and bone resorption-related factors TRAP, MMP-9, and CTsK, while c-Fos controls the expression of CA2 [30]. In this study, DC significantly suppressed the expression of NFATc1 and c-Fos, and these effects appear to be mediated by MAPKs and NF-κB. We subsequently validated the effect of DC on the expression of various bone-related genes controlled by NFATc1 and c-Fos.

Cell–cell fusion is essential for a variety of cell biological processes [31]. Osteoclasts and osteoclast progenitors are unique cells found only in bone, and the process of fusion is initially initiated by the migration/recruitment of progenitor cells. Although osteoclast fusion is not essential for bone resorption, it has a major impact on its efficiency, which is crucial for the size and resorptive capacity of osteoclasts [31]. RANKL-induced osteoclast fusion involves the expression of ATP6v0d2 and DC-STAMP, both of which are tied to the expression of NFATc1 [29]. ATP6V0d2 is a component of the ATPase pump, and ATP6V0d2−/− mice have blocked fusion of progenitor cells to osteoclasts. Interestingly, they do not develop defects in osteoclast activity and differentiation, but a significant increase in osteoblast activity was observed. In 2006, Toshio Suda’s lab used a DNA subtraction screen to show that DC-STAMP is not observed in progenitor cells but is highly expressed in osteoclasts, and by developing DC-STAMP−/− mice, its deficiency induced osteocalcinosis [32]. It was also demonstrated that this occurrence was caused by a defect in giant cells, suggesting that it plays a major role in osteoclast fusion. However, unlike ATP6v0d2, DC-STAMP did not affect the differentiation and activity of osteoblasts [33]. F-actin rings play a major role in the formation of the sealing zone of osteoclasts and are important for bone resorption. Various studies have shown that DC-STAMP plays a major role in this structure, and ATP6v0d2 has been shown to play a major role in the size of the F-actin ring, although it has not been shown to play a role in the number of F-actin rings [34]. In this study, DC strongly suppressed the size and number of F-actin rings, and this was demonstrated to be achieved through suppression of the expression of two genes. 

In the post-maturation bone resorption phase of osteoclasts, the expression of various enzymes and factors plays a role [19]. Firstly, CA2 plays a major role in bone acidification, creating an environment for other enzymes to play an active role in the F-actin ring-enclosed space within the bone, and also creating a suitable environment for osteoclast differentiation [35]. Second, MMP-9 is a protein that plays a key role in the trafficking of osteoclast transcription factors and is an important enzyme that degrades the extracellular matrix [36]. Finally, CTsK is a cysteine protease synthesized and secreted by osteoclasts that plays a role in reducing collagen and matrix proteins during bone resorption, and when its expression is inhibited, bone formation by osteoblasts is promoted [37]. In this study, DC significantly controlled the expression of these genes and consequently reduced the area of pit taken up by osteoclasts through inhibition of bone resorption.

Osteoblasts are mesenchymal cells that play an important role in maintaining bone homeostasis. They are responsible for the production of extracellular matrix proteins, regulating matrix mineralization, controlling bone remodeling, and regulating osteoclast differentiation [38]. For many years, osteoporosis research has focused on inhibiting osteoclast maturation and proliferation, but this does not help reverse already advanced osteoporosis, and may not play a role in osteoblast activity induced by reduced metabolism due to aging or steroids [39]. However, recent advances in bone physiology have led to a number of studies that have identified osteoblast activity as a major target for osteoporosis treatment, along with osteoclast inhibition [38]. The primary role of osteoblasts is to produce a characteristic mixture of extracellular matrix proteins, including type I collagen, the main component of bone, and to maintain homeostasis through regulation of calcium metabolism [38]. The activity of these cells is regulated by two key mechanisms: BMP-2/Smad and Wnt/β-catenin. (i) After binding to the BMP receptor, BMP2 induces phosphorylation of Smad1/5/8 and activates it, and the complex with Smad4 acts as a transcriptional activator and translocates into the nucleus to induce osteoblast differentiation [7]. (ii) The canonical Wnt signaling pathway involving Wnt and β-catenin was found to play a critical role in the regulation of osteoblast differentiation, proliferation, apoptosis, and bone mass in bone tissue and is also required for bone response to mechanical loading [8]. When Wnt binds to Lrp5/6 and Frizzled proteins, the intracellular protein Dishevelled (Dvl) is activated, glycogen synthase kinase-3 (GSK-3beta) is inhibited, and Axin is freed from β-catenin to bind to the cytoplasmic tail of LRP5/6. Subsequently, the β-catenin degradation complex is dissociated and β-catenin accumulates in the cytoplasm, where it enters the nucleus and binds to T-cell factor/lymphoid-enhancer binding factor (TCF/LEF), leading to target gene expression [8]. The activity of these two mechanisms upregulates the expression of RUNX2, which is known to be a key factor in osteoblast differentiation [40]. Overexpression of RUNX2 accelerated osteoblast differentiation, inhibited chondrocyte differentiation, and caused limb defects, and RUNX2 −/− skull cells proliferated faster than wild-type skull cells in vitro [40]. RUNX2 also inhibited the proliferation of osteogenic potential cells and osteosarcoma cells, and transfection of siRNA against RUNX2 into human mesenchymal stem cells increased their proliferation [41]. RUNX2 activity regulates the expression of various proteins during osteoblast differentiation. ALP, a glycoprotein enzyme, is an early differentiation factor in the transition from pre-osteoblasts to mature osteoblasts, and procol1 is required for bone formation [42]. Procol1 is a regulator of osteoblast differentiation that promotes bone formation [43]. The transition from mature osteoblast to osteocyte involves the secretion of extracellular matrix proteins such as OPN and OCN, which mineralize the matrix of the osteocyte [44]. In this study, DC promoted the calcification of osteoblast nodules, and the effect was attributed to the role of various osteogenesis-related factors induced through the upregulation of BMP-2 and Wnt mechanisms. This suggests that DC is not just a drug with a mechanism for treating osteoporosis through osteoclast inhibition, but also has the potential to repair damaged bone. A schematic diagram of the mechanism of osteoclasts and osteoblasts in bone metabolism of DC is shown in Figure 7.

LPS is a typical Gram-negative bacterial endotoxin and is a key factor in the development of inflammatory bone loss. LPS stimulates NF-κB activation, followed by the release of inflammatory cytokines, including TNF-α and IL-1, which induce osteoclast differentiation and function, resulting in osteoporosis [45]. In the present study, a strong decrease in bone mineral density was observed in the LPS-treated group compared to the normal group of mice, and DC inhibited this decrease. In addition, DC improved not only bone density but also the quality of various bone microarchitectures, such as the separation, number, and thickness of trabeculae. Taken together, the results from both cellular and animal studies demonstrate that DC are effective at all stages of osteoclast differentiation, fusion, and maturation, and can significantly inhibit the development of osteoporosis in vivo. 

These results are a validation of the initial hypothesis that DC may be effective in bone metabolism due to their anti-inflammatory and antioxidant effects, and DC may represent a new alternative in the osteoporosis market, which is currently in need of new treatment alternatives due to their serious side effects. In addition, it is not limited to osteoporosis per se, but may have applications in various bone metabolic diseases caused by disorders of osteoclasts and osteoblasts. The limitations of this study include the following: (i) This study only investigated the role of DC in inflammatory osteoporosis in mice. Osteoporosis is caused by various etiological factors such as menopause, steroids, and aging, and each disease is induced through different mechanisms [46]. Therefore, it would be of great research value to investigate the role of DC in bone disease through experimental studies in various osteoporosis models. (ii) We assumed that DC would have fewer side effects because it is a natural product-based extract, and no side effects were observed in cell and animal experiments. However, independent studies (single- and repeated-dose toxicity studies, pharmacokinetics, etc.) should be conducted to confirm this [47]. (iii) Among the various factors that assess bone microarchitecture, the ellipsoid factor (EF) measures the extent to which the microarchitecture of trabecular bone is shaped and unshaped, indicating whether it is densely aligned [48]. It is a key factor in understanding the health of bone tissue. However, this factor was not analyzed in this study. Therefore, the actual improvement in bone quality along with the change in bone density remains questionable. In the future, if this indicator is studied through further analysis, it would be beneficial to prove the anti-osteoporosis effect of DC.

## 4. Materials and Methods

### 4.1. Reagents

Dulbecco’s Modified Eagle’s Medium (DMEM) and DPBS were purchased from WELGENE (Daejeon, Republic of Korea), and α-Minimal Essential Medium (αMEM) and penicillin/streptomycin (p/s) were purchased from gibco (Gaithersburg, MD, USA). Fetal bovine serum (FBS) was purchased from Atlas Biologicals (Fort Collins, CO, USA). RANKL (cat. 315-11-100) was obtained from Peprotech (London, UK). DC (cat. CFN98713) was obtained from ChemFaces (Wuhan, China). The cell counting kit (CCK-8) kit was supplied by Dojindo Molecular Technologies (Kumamoto, Japan). Ascorbic acid, β-glycerophosphate (B.G.P.), dimethyl sulfoxide (DMSO), TRAP assay kit, DAPI, and LPS (cat. L-2880) were purchased from Sigma-Aldrich (St. Louis, MO, USA). Acti-stain™ 488 Fluorescent Phalloidin was obtained from Cytoskeleton, Inc. (Denver, CO, USA). Secondary antibodies were purchased from Jackson ImmunoResearch Laboratories, Inc. (West Grove, PA, USA). RNAiso Plus was purchased from TaKaRa (Kusatsu, Japan), and chloroform and isopropyl alcohol were purchased from DAEJUNG (Siheung-si, Republic of Korea). Ethyl alcohol was supplied by SAMCHUN (Pyeongtaek-si, Republic of Korea). Agaros was purchased from LONZA (Rockland, ME, USA), and SYBR Safe DNA Gel Stain was purchased from Invitrogen ( Burlington, ON, Canada; Carlsbad, CA, USA). The primers used were obtained from GENOTECH (Daejeon, Republic of Korea).

### 4.2. Cell Culture

RAW 264.7, a murine macrophage cell line, was acquired from the Korea Cell Line Bank (Seoul, Republic of Korea). These cells were cultivated in DMEM supplemented with 10% FBS and 100 units/mL of penicillin/streptomycin. MC3T3-E1, a pre-osteoblast cell line, was procured from the American Type Culture Collection (ATCC, Washington, DC, USA). The cells were cultured in αMEM (without ascorbic acid) supplemented with 10% FBS and 100 units/mL of penicillin/streptomycin. The cell cultures were maintained in a 37 °C environment with 5% CO_2_ using a Thermo Fisher Scientific cell culture incubator (Waltham, MA, USA).

### 4.3. Cell Viability Measurement

The concentration of DC used in this study was set by referring to previous studies. In previous studies, DC was tested up to 200 µg/mL (346 µM), and a concentration of 100 µM was universally set as the highest concentration. Therefore, we set the concentrations for this experiment at 50, 100, and 200 µg/mL (86, 173, and 346 µM) and verified their intracellular toxicity [12,49,50]. RAW 264.7 cells were seeded onto a 96-well plate at a density of 5 × 10^3^ cells/100 µL. The cells were then incubated for 24 h in a 37 °C environment with 5% CO_2_. Subsequently, the fermentation product was applied and treated for an additional 24 h. MC3T3-E1 were seeded onto a 96-well plate at a density of 5 × 10^4^ cells/100 µL. The cells were then incubated for 24 h and 72 h in a 37 °C environment with 5% CO_2_. Subsequently, the fermentation product was applied and treated for an additional 24 h. Following the treatment, CCK-8 reagent was added, and the cells were further incubated for 2 h. Absorbance measurements were obtained at a wavelength of 450 nm using a microplate reader.

### 4.4. Osteoclastogenesis and TRAP Assay

In order to induce the differentiation of RAW 264.7 cells into osteoclasts, an initial seeding of RAW 264.7 cells was performed, with a density of 1.5 × 10^4^ cells in 400 µL within each well of a 24-well plate. Subsequently, the cells were maintained in αMEM supplemented with 100 ng/mL RANKL for a period of 5 days. The same culture medium was exchanged every 2 days. Afterward, the cells underwent TRAP staining. They were fixed in 4% formalin for 10 min at RT and then treated with a TRAP staining assay kit for 1 h at 37 °C. Cells with three or more nuclei that stained positive for TRAP were counted as osteoclasts, and images of the osteoclasts were captured using an inverted microscope (Olympus, Tokyo, Japan). TRAP activity was then transferred to a new 96-well plate, and the reaction took place in an incubator at 37 °C for 1 h. This reaction included equal amounts of medium and TRAP solution (4.93 mg p-nitrophenyl phosphate (Pnpp) in 750 μL of 0.5 M acetate solution and 150 μL of tartrate acid solution). After adding 0.5 M NaOH to the reaction plate, we measured the absorbance at 405 nm using a microplate reader (VersaMax microplate reader, Molecular Devices, San Jose, CA, USA).

### 4.5. F-Actin Ring and Pit Formation Assay

To measure F-actin ring formation in DC, F-actin ring immunofluorescence assay was performed. RAW 264.7 cells were seeded in a 96-well plate at 5 × 10^3^ cells/well and cultured for 24 h. Afterwards, they were treated with RANKL (100 ng/mL) and DC (50, 100 and 200 μg/mL) and cultured at 37 °C for 5 days. Cells that had completed differentiation were fixed with 4% formalin for 20 min, and cell permeability was increased by reacting with 0.1% Triton X-100 (in PBS) for 5 min. Cells were stained using Acti-stain™ 488 Fluorescent Phalloidin in a light-blocked space for 30 min. Then, the cells were counterstained with DAPI. After treatment with DAPI, images were captured using a fluorescence microscope (Cellena; Logosbio, Anyang, Republic of Korea), and the number of stained cells was counted.

To determine the inhibitory effect of DC on osteoclast-formed pit formation, RAW 264.7 cells were seeded in an osteo-coated plate (Corning, NY, USA) at 5 × 10^3^ cells/well and cultured for 24 h. Afterwards, they were treated with RANKL (100 ng/mL) and DC (50, 100, and 200 μg/mL) and cultured at 37 °C for 5 days. The differentiated cells were lysed with NaClO and dried thoroughly, and a total of 6 spots were photographed under an inverted microscope at a field of view of X100. Data were expressed as the percentage of area taken up by osteoclasts relative to the total area.

### 4.6. Western Blot

RAW 264.7 cells were seeded in 60 π dishes at a density of 2 × 10^6^, 5 × 10^5^ cells/well, respectively, according to the confirmation of MAPK signaling pathway expression and NFATc1/c-Fos expression, and cultured for 24 h. To confirm the expression of the MAPK signaling pathway, αMEM was treated with DC (50, 100 and 200 μg/mL) for 6 h and then stimulated with RANKL (100 ng/mL) for 30 min. To confirm NFATc1/c-Fos expression, αMEM containing RANKL (100 ng/mL) was treated with DC (50, 100 and 200 μg/mL) for 24 h. To confirm the expression of transcription factors for osteoblast differentiation, MC3T3-E1 cells were seeded in a 60 π plate at a density of 5 × 10^5^ cells/well and cultured for 24 h. Then, the cells were treated with DC (50, 100, and 200 μg/mL) for 2 days. All cells used were cultured in a 5% CO_2_ incubator environment at 37 °C, and the culture medium was changed every 2 days. Total protein was homogenized in RIPA buffer (composition: 50 mM Tris-Cl, 150 mM NaCl, 1% NP-40, 0.5% sodium deoxycholateand, 0.1% SDS). Homogenates were centrifuged at 13,200 rpm for 20 min at 4 °C, and the supernatant was gathered for Western blotting analysis. Protein (30 ug) quantified with BCA protein analysis was separated by SDS-PAGE (10% acrylamide) and transferred to nitrocellulose membranes. After that, the membrane was blocked using 5% skim milk in TBST and incubated with a primary antibody overnight at 4 °C. The primary antibody used, p-ERK (cat. No.: 4370S; 1:1000), ERK (cat. No.: 4695S; 1:1000), p-p38 (cat. No.: 4511S; 1:1000), p38 (cat. No.: 9212L; 1:1000), NFATc1 (cat. No.: 556602; 1:1000), c-Fos (cat. No.: sc-447; 1:200), BMP-2 (cat. No.: ab14933; 1:1000), Osterix (cat. No.: ab209484; 1:1000), RUNX2 (cat. No.: ab; 1:1000), p-Smad (cat. No.: 9516S; 1:1000), Smad (cat. No.: ab80255; 1:500), and actin (cat. No.: 8432; 1:1000), was incubated overnight at 4 °C. After washing, membranes were incubated with horseradish peroxidase-conjugated secondary antibody for 2 h at RT. Following incubation, membranes were washed and developed using a electrochemiluminescence (ECL) solution. The housekeeping protein actin was used for normalization.

### 4.7. RT-PCR

RAW 264.7 cells were seeded in a 6-well plate at a density of 2 × 10^5^ cells/well and cultured for 24 h. For the next 4 days, RANKL (100 ng/mL) and DC (50, 100 and 200 μg/mL) were treated. To confirm the expression of transcription factors for osteoblast differentiation, MC3T3-E1 cells were seeded in a 6-well plate at a density of 5 × 10^5^ cells/well and cultured for 24 h. Then, the cells were treated with DC (50, 100, and 200 μg/mL) for 4 days. RNA obtained by extraction using TRIzol reagent was quantified using Nano drop (Thermo, Waltham, MA, USA) and reverse transcribed into cDNA (PCR machine, BIO-RAD, Hoboken, NJ, USA). The synthesized cDNA was amplified using the KAPA Taq Extra PCR Kit. The conditions for each indicator and the primer sequences are listed in Table 1. The PCR product was electrophoresed by adding SYBR (1:10,000) to a 2% agarose gel, and the gene was confirmed using gel doc. GAPDH was used as the housekeeping gene.

### 4.8. Osteoblast Differentiation and Alizarin Red S Staining

In order to induce the differentiation of MC3T3-E1 cells into osteoblasts, an initial seeding of MC3T3-E1 cells was performed, with a density of 1.5 × 10^4^ cells in 700 µL within each well of a 24-well plate. Subsequently, the cells were cultured in differentiation medium (D.M, αMEM (without ascorbic acid) + 10% FBS + 1% P/S + 25 µg/mL ascorbic acid + 10 mM beta-glycerophosphate) for a period of 14 and 17 days. The same culture medium was exchanged every 3 days. When calcification nodules were observed on the plate, they were washed three times with DPBS and fixed by reacting 80% Et-OH at 4 °C for 1 h. Then, the cells were stained with Alizarin red S solution for 20 min, and the plate was observed under an inverted microscope (×200, scale bar: 100 µm). The stained dye was extracted with 10% (*v*/*w*) cetylpyridinium chloride (Sigma-Aldrich, St. Louis, MO, USA) for 15 min, and then measured at the absorbance at 570 nm using an ELISA instrument.

### 4.9. Establishment of an Inflammatory Osteoporosis Model and Identification of the Anti-Osteoporosis Effect of DC

All animal-related processes and management were conducted in accordance with the guidelines of the Kyung Hee University Animal Committee and animal testing was performed [permission number: KHSASP-23-316]. A total of 18 Institute of Cancer Research (ICR) CD-1 mice (male; age, 4 weeks; weighing, 27–29 g) were purchased from Nara Biotech. An environment of 22 ± 2 °C, 12 h light/12 h dark cycle, and relative humidity of 53–55% was provided, and the animals were raised with free access to water and feed. In the animal experiment, all mice underwent an adaptation period for one week before LPS administration. The animals were divided into three groups of six. The animals were divided into three groups: (i) normal (normal saline intraperitoneally), (ii) LPS (5 mg/kg intraperitoneally), and (iii) DC (5 mg/kg of LPS and 20 mg/kg of DC intraperitoneally). LPS was administered twice on days 1 and 4, and DC was administered for a total of 9 days. At the end of treatment, mice were deeply anaesthetized using 100% oxygen and 5% isoflurane, a lethal volume of blood (>800 μL) was extracted via cardiac puncture, and the mice were sacrificed by cervical dislocation. For bone density analysis, the right femur was fixed in 10% neutral buffered formalin (NBF). Afterward, they were washed in running water for 24 h.

### 4.10. Micro-CT Analysis

The femoral head was scanned with micro-CT (SkyScan 1176; Bruker Corporation; Kontich, Antwerpen, Belgium) and analyzed using NRecon software (SkyScan version 1.6.10.1, Bruker Corporation). The X-ray generator parameters were configured as follows: 50 kV/200 μA, 8.9 μm pixel size, an aluminum (Al) filter with a thickness of 0.5 mm, and a 180° rotation angle with a rotation step of 0.4°. Analysis of trabecular lesions in the femur was performed at the patellar surface, extending 0.8 mm proximally from the end of the growth plate, where the trabecular bone featured an offset of 0.1 mm and a height of 0.8 mm. The total number of slides filmed was 200. Bone microstructural parameters such as BV/TV, Tb.Th, Tb.Sp, Tb.N, and SMI were analyzed using mouse femur samples.

### 4.11. Statistical Analysis 

All results were expressed as mean ± S.E.M. Each experiment was repeated at least three times. The significance of the experimental results was tested using one-way ANOVA, followed by Tukey’s post hoc test. A *p* value of 0.05 or less was considered significant, and statistics were performed using GraphPad Prism 9.

## Figures and Tables

**Figure 1 ijms-24-16465-f001:**
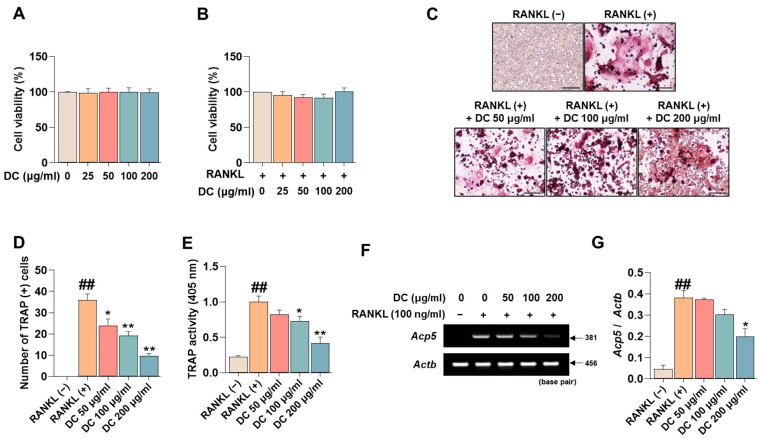
RANKL-induced osteoclast differentiation and expression of the specific marker TRAP is inhibited by expression of DC. (**A**) The cytotoxicity of DC against RAW 264.7 cells and (**B**) osteoclasts was validated by CCK-8 assay. (**C**) The effect of inhibiting the differentiation of osteoclasts was verified by TRAP assay kit (×100, scale bar: 200 µm). (**D**) The number of differentiated osteoclasts was counted using ImageJ software (version 1.46). (**E**) The activity of TRAP in the culture medium was verified by TRAP solution and ELISA instrument. (**F**) The mRNA expression of TRAP (*Acp5*) was validated by RT-PCR. (**G**) The expression of the TRAP was quantified by β-actin (*Actb*). All experiments were repeated at least three times. Data were expressed as mean ± SEM. ## *p* < 0.01 versus the non-treated cells and ** *p* < 0.01 and * *p* < 0.05 versus the RANKL-treated cells.

**Figure 2 ijms-24-16465-f002:**
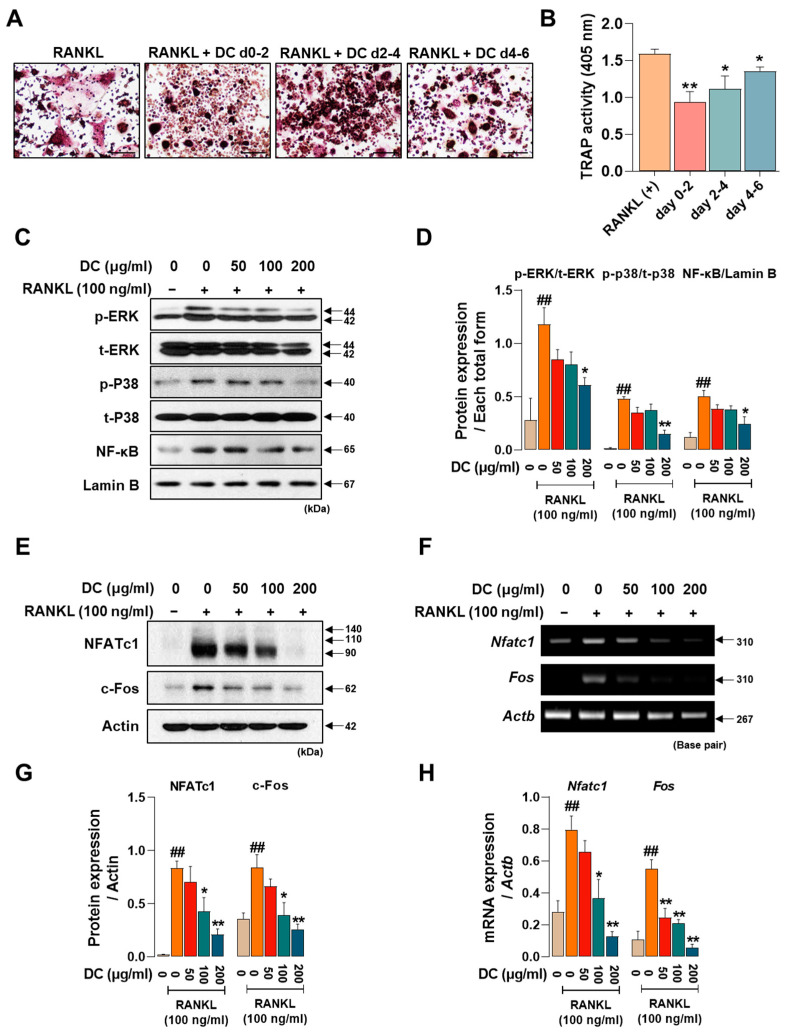
DC exhibited a significant role in modulating the initial stages of osteoclast differentiation by suppressing the expression of NFATc1 and c-Fos through the regulation of MAPKs phosphorylation and the inhibition of NF-κB nuclear translocation. (**A**) TRAP staining was performed after drug treatment at different time points to validate DC during osteoclast differentiation (×100, scale bar: 200 µm). (**B**) The activity of TRAP in the culture medium was validated by TRAP solution and ELISA instrument. (**C**) The effects of DC on the phosphorylation of MAPKs and nuclear translocation of NF-κB were validated by Western blot. (**D**) The expression of factors was quantified using their respective total forms or Lamin B. (**E**) The protein and (**F**) mRNA expression of osteoclast essential transcription factors NFATc1 and c-Fos were verified by Western blot and RT-PCR. (**G**,**H**) Expression of factors was measured via ImageJ software (version 1.46) and quantified via β-actin (*Actb*). All experiments were repeated at least three times. Data were expressed as mean ± SEM. ## *p* < 0.01 versus the non-treated cells and ** *p* < 0.01 and * *p* < 0.05 versus the RANKL-treated cells.

**Figure 3 ijms-24-16465-f003:**
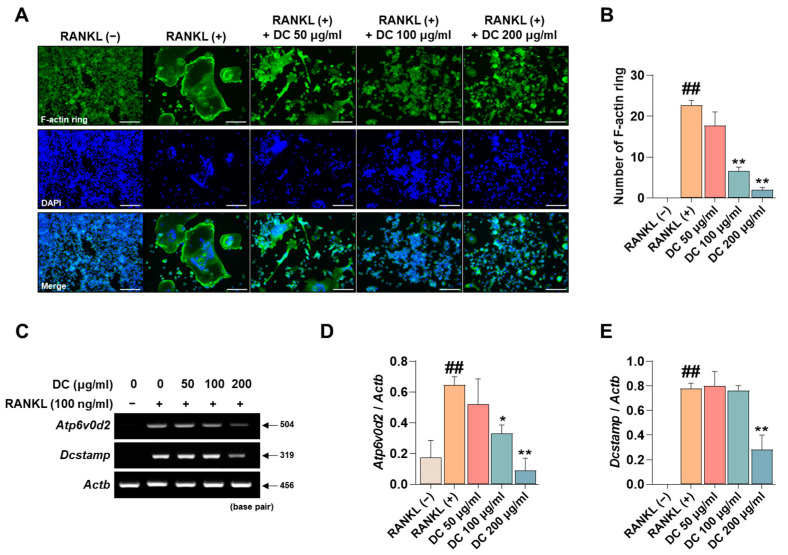
DC regulated the formation of F-actin rings through inhibition of *Atp6v0d2* and *Dcstamp* expression. (**A**) F-actin rings formed by RANKL were imaged by fluorescence microscopy with the Acti-stain 488 phalloidin (×100, scale bar: 200 µm). (**B**) The number of F-actin rings was counted using ImageJ software (version 1.46). (**C**) The mRNA expression of *Atp6v0d2* and *Dcstamp* was verified by RT-PCR. (**D**) The expression of *Atp6v0d2* and (**E**) *Dcstamp* was measured by ImageJ software (version 1.46) and quantified as β-actin (*Actb*). All experiments were repeated at least three times. Data were expressed as mean ± SEM. ## *p* < 0.01 versus the non-treated cells and ** *p* < 0.01 and * *p* < 0.05 versus the RANKL-treated cells.

**Figure 4 ijms-24-16465-f004:**
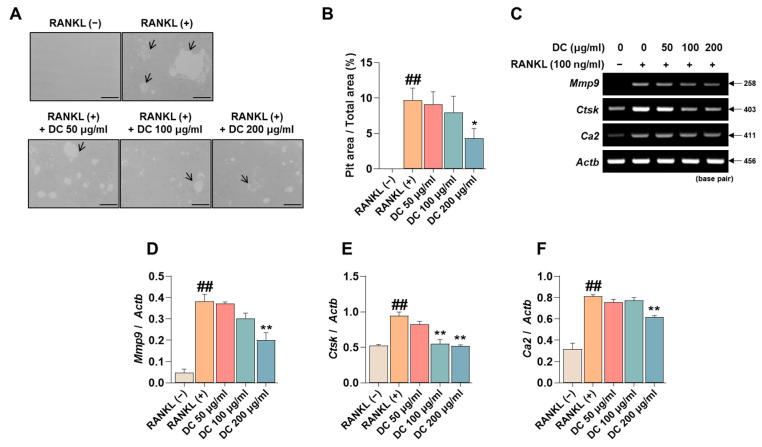
DC inhibited pit formation by controlling bone resorption gene expression. (**A**) To evaluate the effectiveness of DC on osteoclast bone resorption, an osteo assay kit and inverted microscopy were used (×100, scale bar: 200 µm). The area absorbed by osteoclasts is indicated by a black arrow. (**B**) The area of the pit was measured by ImageJ. (**C**) The expression of bone resorption-related genes (*Mmp9*, *Ctsk*, *Ca2*) was validated by RT-PCR. (**D**) The expression of MMP-9, (**E**) CTsK, and (**F**) CA2 was measured using ImageJ software (version 1.46) and quantified by β-actin (*Actb*). All experiments were repeated at least three times. Data were expressed as mean ± SEM. ## *p* < 0.01 versus the non-treated cells and ** *p* < 0.01 versus the RANKL-treated cells. * *p* < 0.05.

**Figure 5 ijms-24-16465-f005:**
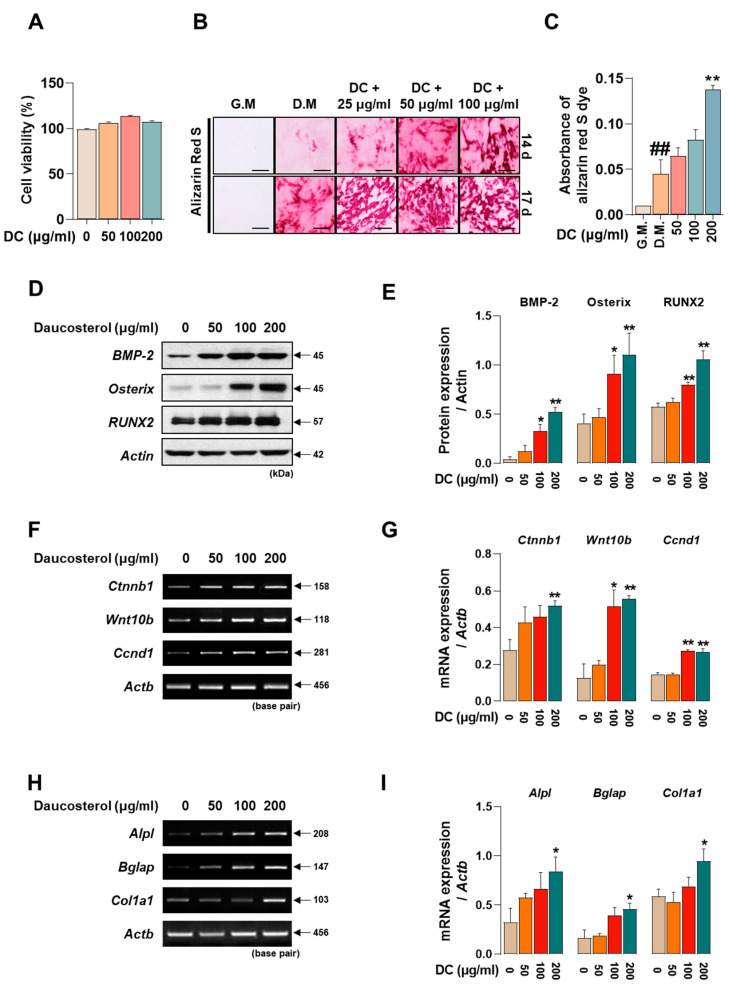
DC promoted osteoblast differentiation and calcified nodule formation through BMP-2/Smad and Wnt/β-catenin mechanisms. (**A**) The effect of DC on the cytotoxicity of MC3T3-E1 was validated by CCK-8 assay. (**B**) The effect of DC on calcification nodule formation was validated by Alizarin red S staining (×200, scale bar: 100 µm). (**C**) The stained Alizarin red S dye was extracted and the absorbance was measured by ELISA instrument. (**D**) The effect of DC on BMP-2/Smad mechanism was validated by Western blot. (**E**) The expression of factors was measured by ImageJ and quantified by β-actin. (**F**) The validity of DC for the Wnt/β-catenin mechanism was validated by RT-PCR. (**G**) The expression of factors was measured by ImageJ software (version 1.46) and quantified by *Actb*. Data were expressed as mean ± SEM. (**H**) The validity of DC for the osteoblast-related genes was validated by RT-PCR. (**I**) The expression of factors was measured by ImageJ software (version 1.46) and quantified by *Actb*. Data were expressed as mean ± SEM. ## *p* < 0.01 versus the growth medium (G.M)-cultured cells and ** *p* < 0.01 and * *p* < 0.05 versus the differentiation medium (D.M)-cultured cells.

**Figure 6 ijms-24-16465-f006:**
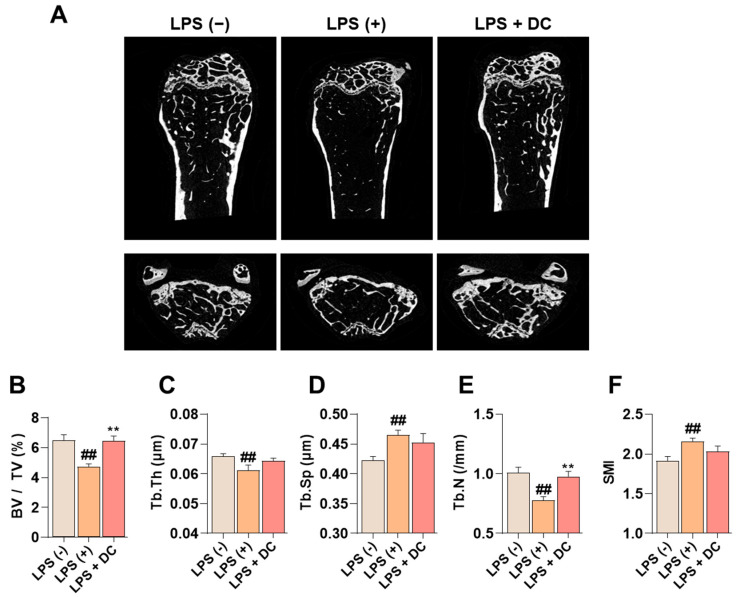
DC inhibits the decrease in bone mineral density in the femur of LPS-induced osteoporotic mice. (**A**) Bone density reduction with LPS administration was visualized by micro-CT imaging. (**B**) Bone microarchitecture indicators BMD, (**C**) Tb.Th, (**D**) Tb.N, (**E**) Tb.Sp, and (**F**) SMI were measured by micro-CT analysis software. There were six mice in each group. Data were expressed as mean ± SEM. ## *p* < 0.01 versus the non-treated mice and ** *p* < 0.01 versus LPS-induced mice.

**Figure 7 ijms-24-16465-f007:**
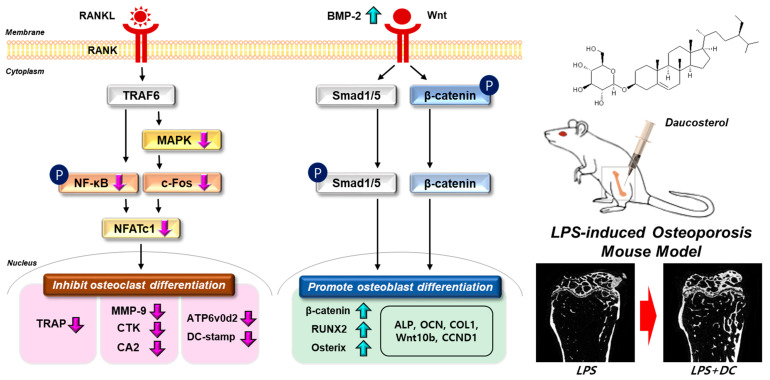
Schematic representation of the validity of DC on bone metabolism. The signaling system in each cell is indicated by a black arrow. Changes in bone density through administration of DC are indicated by red arrows. Phosphorylation of the indicator is indicated as P.

**Table 1 ijms-24-16465-t001:** Primer sequences used for RT-PCR.

Cells (Source)	Gene Name	Sequence (5′-3′)	Accession No.	Tm (°C)	Base Pair
Osteoclast (mouse)	*Acp5* (TRAP)	F: ACT TCC CCA GCC CTT ACT ACC G R: TCA GCA CAT AGC CCA CAC CG	NM_007388.3	58	381
	*Nfatc1* (NFATc1)	F: TGC TCC TCC TCC TGC TGC TC R: CGT CTT CCA CCT CCA CGT CG	NM_198429.2	58	480
	*Fos* (c-Fos)	F: ATG GGC TCT CCT GTC AAC AC R: GGC TGC CAA AAT AAA CTC CA	NM_010234.3	58	480
	*Atp6v0d2* (ATP6v0d2)	F: ATG GGG CCT TGC AAA AGA AAT CTG R: CGA CAG CGT CAA ACA AAG GCT TGT A	NM_175406.3	64	504
	*Dcstamp* (DC-STAMP)	F: TGG AAG TTC ACT TGA AAC TAC GTG R: CTC GGT TTC CCG TCA GCC TCT CTC	NM_001289506.1	63	319
	*Mmp-9* (MMP-9)	F: CGA CTT TTG TGG TCT TCC CC R: TGA AGG TTT GGA ATC GAC CC	NM_013599.4	58	258
	*Ctsk* (CTsK)	F: AGG CGG CTA TAT GAC CAC TG R: CCG AGC CAA GAG AGC ATA TC	NM_007802.4	58	403
	*Ca2* (CA2)	F: CTC TCA GGA CAA TGC AGT GCT GA R: ATC CAG GTC ACA CAT TCC AGC A	NM_001357334.1	58	411
	*Actb* (β-actin)	F: TTC TAC AAT GAG CTG CGT GT R: CTC ATA GCT CTT CTC CAG GG	NM_007393	58	456
Osteoblast (mouse)	*Ctnnb1* (β-catenin)	F: TGC TGA AGG TGC TGT CTG TC R: CTG CTT AGT CGC TGC ATC TG	NM_001165902.1	59	158
	*Wnt10b* (Wnt10b)	F: TTC TCT CGG GAT TTC TTG GAT TC R: TGC ACT TCC GCT TCA GGT TTT C	NM_011718.2	59	118
	*Ccnd1* (Ccnd1)	F: GAA GGA GAT TGT GCC ATC R: TTC TTC AAG GGC TCC AGG	[51]	55	346
	*Alpl* (ALP)	F: CGG GAC TGG TAC TCG GAT AA R: TGA GAT CCA GGC CAT CTA GC	NM_001287172.1	55	208
	*Bglap2* (OCN)	F: GCA ATA AGG TAG TGA ACA GAC TCC R: GTT TGT AGG CGG TCT TCA AGC	NM_001032298.3	59	147
	*Col1a1* (COL1)	F: GCT CCT CTT AGG GGC CAC T R: CCA CGT CTC ACC ATT GGG G	NM_007742.4	60	103

## Data Availability

Data are contained within the article.

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
