# Peer review of "Exploring the Anti-Osteoporotic Potential of Daucosterol: Impact on Osteoclast and Osteoblast Activities"

_ijms, 2023, doi:10.3390/ijms242216465_

Round 1
Reviewer 1 Report
Comments and Suggestions for Authors
This is a very interesting paper that is well written and presented. I have only a few comments.
Introduction
The inclusion of more brief detail on the potential mechanism of action of daucosterol would strengthen the paper. It would aid the reader to better understand why it possesses anti-cancer, anti-hepatic injury etc. properties. Further speculation on it may be important for the treatment of osteoporosis is also warranted. Following on from this, the inclusion of a hypothesis would also strengthen the manuscript.
Methods
How were the doses chosen? Please clarify why only 50, 100, 200 uM were investigated in osteoclasts.
Similar to the comment above, and also in the discussion section, please do speculate on what properties of daucosterol may cause this response (e.g., scavenging of ROS?). Comparison of these results to other non-bone focused studies may be helpful.
Author Response
Responses to Reviewers:
We would like to thank both of the reviewers for their positive and very helpful comments to improve our manuscript. In response to reviewers, we have addressed the comments as follow:
Reviewer 1.
This is a very interesting paper that is well written and presented. I have only a few comments.
- We have incorporated the reviewer's comments and improved the manuscript. We appreciate the developmental comments. We have highlighted the revisions in the manuscript.
Introduction
The inclusion of more brief detail on the potential mechanism of action of daucosterol would strengthen the paper. It would aid the reader to better understand why it possesses anti-cancer, anti-hepatic injury etc. properties. Further speculation on it may be important for the treatment of osteoporosis is also warranted. Following on from this, the inclusion of a hypothesis would also strengthen the manuscript.
- We wrote in the introduction that we hypothesize that DC may play a positive role in bone metabolism. This is because the anti-inflammatory and antioxidant effects of DC may show the potential to have a positive impact on bone metabolism. Please review this statement.
- The sentence we added is as follows: Various recent studies have shown that natural compounds with anti-inflammatory effects may be effective against osteoclasts and osteoporosis through inhibition of the expression of RANKL and inflammatory cytokines [14,15]. In addition, inhibition of reactive oxygen species (ROS) through antioxidant action inhibits the activity of osteoclasts and activates osteoblasts [16]. For these reasons, the various pharmacological effects demonstrated in previous studies of DC were expected to make them suitable as an effective drug for bone metabolism.
Methods
How were the doses chosen? Please clarify why only 50, 100, 200 uM were investigated in osteoclasts.
- We checked the cell treatment concentration by referring to the previous daucosterol paper (references are at the bottom). Here are the results we found [experimental concentrations: 50, 100, 200 ug/ml (MW: 576.859, Molar concentration: 86, 173, 346 uM)]. We found that the DC at 100 uM were universally tested, with the highest concentration being 200 ug/ml (346 uM). We then confirmed that the CCK-8 data for each cell showed no toxicity, so we ran the experiment at that concentration. We added that to the method.
- Daucosterol Inhibits the Proliferation, Migration, and Invasion of Hepatocellular Carcinoma Cells via Wnt/β-Catenin Signaling
- Daucosterol inhibits cancer cell proliferation by inducing autophagy through reactive oxygen species-dependent manner
- Daucosterol inhibits colon cancer growth by inducing apoptosis, inhibiting cell migration and invasion and targeting caspase signalling pathway
Similar to the comment above, and also in the discussion section, please do speculate on what properties of daucosterol may cause this response (e.g., scavenging of ROS?). Comparison of these results to other non-bone focused studies may be helpful.
- We have added a discussion of the hypotheses mentioned in the introduction to the end of the discussion and split the paragraphs to make the conclusion stand out better. We really appreciate your developmental review.
- The revised sentence reads as follows: These results are a validation of the initial hypothesis that DC may be effective in bone metabolism due to their anti-inflammatory and antioxidant effects, and DC may represent a new alternative to the osteoporosis market, which is currently in need of new treatment alternatives due to their serious side effects.
Reviewer 2 Report
Comments and Suggestions for Authors
The agent daucosterol (DC) is present in many families of medicinal plants, has anti-cancer effects and is considered as a promising drug against osteoporosis. In the present study, Lee and coworkers demonstrated that DC inhibited the initial stages of osteoclast differentiation, thus preventing the formation of mature multinucleated osteoclasts. In addition, DC promoted osteoblast differentiation. In mice in which osteoporosis was induced by treatment with lipopolysaccharide (LPS), DC inhibited bone density loss.
These results are not new and have already been published by Jae-Hyun Kim et al. 2019 (Leonurus sibiricus L. ethanol extract promotes osteoblast differentiation and inhibits osteoclast formation). Moreover, the same experiments have been performed in the present study. Curiously, Lee et al. do not mention the study by Kim et al. although three authors of the present study were involved in it.
Major comments
DC has been extracted from two species of the family Lamiaceae (El Omari et al. 2022) and Leonurus sibiricus L. belongs to this family. So, what distinguishes DC from the Leonurus sibiricus extract and what is the novelty and scientific progress of the present findings?
The structural model index (SMI) is used to determine rods and plates geometry in trabecular bone. It uses the change in surface curvature that occurs when a structure varies from spherical to cylindrical to planar.
How did the authors characterize/measure the plate-likeness of the bone? Why did they not use the ellipsoid factor (EF) which seemed to be more reliable for measurements of rods and plates in the trabecular bone?
Minor comments
Did the authors measure the size of osteoclasts after treatment with daucosterol?
Do the authors know if DC treated osteoclasts show a ruffled border?
Fig. 4B. The pit formation assay is not well explained. What did the authors measure? Area of resorption?
Page (P) 8, line (L) 225. It should read decrease
P 12, L 386.The authors did not provide any data, which show that the length of the bony trabeculae changes after DC treatment.
Author Response
Responses to Reviewers:
We would like to thank both of the reviewers for their positive and very helpful comments to improve our manuscript. In response to reviewers, we have addressed the comments as follow:
Reviewer 2.
The agent daucosterol (DC) is present in many families of medicinal plants, has anti-cancer effects and is considered as a promising drug against osteoporosis. In the present study, Lee and coworkers demonstrated that DC inhibited the initial stages of osteoclast differentiation, thus preventing the formation of mature multinucleated osteoclasts. In addition, DC promoted osteoblast differentiation. In mice in which osteoporosis was induced by treatment with lipopolysaccharide (LPS), DC inhibited bone density loss.
These results are not new and have already been published by Jae-Hyun Kim et al. 2019 (Leonurus sibiricus L. ethanol extract promotes osteoblast differentiation and inhibits osteoclast formation). Moreover, the same experiments have been performed in the present study. Curiously, Lee et al. do not mention the study by Kim et al. although three authors of the present study were involved in it.
- We really appreciate your constructive/improvemental review. We have carefully reviewed and revised our manuscript in light of your comments. Thank you again for taking the time to review our manuscript.
Major comments
DC has been extracted from two species of the family Lamiaceae (El Omari et al. 2022) and Leonurus sibiricus L. belongs to this family. So, what distinguishes DC from the Leonurus sibiricus extract and what is the novelty and scientific progress of the present findings?
- First of all, we would like to mention that Leonurus sibiricus L. and daucosterol are completely different. Leonurus sibiricus L. is extracted by cold soaking in ethanol and contains a variety of active substances in the sample. On the other hand, daucosterol is a single constituent. Also, in the reviewer's comment, you mentioned that daucosterol is detected in the family Lamiaceae, but there is no study showing the presence of daucosterol in Leonurus sibiricus L. If you refer to the following manuscript, you will find that Leonurus sibiricus L. contains a variety of constituents, but daucosterol is not present. (Leonurus cardiaca L. (Motherwort): A Review of its Phytochemistry and Pharmacology). Therefore, the two studies cannot be compared We clarify that we did not reference their manuscript because the two studies are clearly different, and we did not intentionally omit it.
- Regarding the similarity of the study design, validating the activity of osteoclasts and osteoblasts in osteoporotic diseases and explaining their mechanisms are already utilized in various studies to validate drugs and have already been proven to be suitable to be used as therapeutic targets for the disease. In addition, inducing osteoporosis through LPS injection and validating the anti-osteoporotic effect in femur are also used in many studies. Therefore, we do not believe it is correct to say that our study is problematic simply because of the similarity of the study design to the Kim et al. (2019) study.
The structural model index (SMI) is used to determine rods and plates geometry in trabecular bone. It uses the change in surface curvature that occurs when a structure varies from spherical to cylindrical to planar.
- We calculated the SMI measurements using an analysis program after the micro-CT acquisition. We identified that this was missing from the Materials and Methods section, added it, and provided detailed acquisition conditions for the micro-CT analysis.
How did the authors characterize/measure the plate-likeness of the bone? Why did they not use the ellipsoid factor (EF) which seemed to be more reliable for measurements of rods and plates in the trabecular bone?
- EF was not measurable in our analysis program. We agree with the reviewer that EF is a valid method to verify the alignment of trabeculae within bone tissue. Therefore, we have listed our inability to analyze it as a limitation of the study in the Discussion section.
- The added sentences are: iii) Among the various factors that assess bone microarchitecture, the ellipsoid factor (EF) measures the extent to which the microarchitecture of trabecular bone is shaped and unshaped, indicating whether it is densely aligned. It is a key factor in understanding the health of bone tissue. However, this factor was not analyzed in this study. Therefore, the actual improvement in bone quality along with the change in bone density remains questionable. In the future, if this indicator is studied through further analysis, it would be beneficial to prove the anti-osteoporosis effect of DC.
Minor comments
Did the authors measure the size of osteoclasts after treatment with daucosterol?
- We did not measure the size of the osteoclasts; our study objective was the effect of DC on osteoclast numbers, and we counted multinucleated, giant, and red (TRAP-positive) cells. The criterion for giant cells was larger cells compared to the untreated RAW 264.7 cells. In the RANKL-treated group, we found that most of the cells fused into giant cells.
Do the authors know if DC treated osteoclasts show a ruffled border?
- Although the role of DC in the ruffled border was not directly confirmed in this study, we demonstrated that F-actin structures are critical for the formation and maintenance of the ruffled border.
Fig. 4B. The pit formation assay is not well explained. What did the authors measure? Area of resorption?
- We recognized the lack of data on pit formation and added it. We photographed 6 spots at a field of view of 100x through the microscope and then expressed the area of absorbed pits as a percentage of the total area. This has been added to the Materials and methods section.
Page (P) 8, line (L) 225. It should read decrease
- We corrected increased to inhibited in that sentence. We very much appreciate the reviewer's detailed point.
P 12, L 386.The authors did not provide any data, which show that the length of the bony trabeculae changes after DC treatment.
- We realized that we had mistakenly written thickness for length and corrected it. We also corrected area to separation in that sentence. We added more detail to the micro-CT analysis method.
Round 2
Reviewer 2 Report
Comments and Suggestions for Authors
In the revised version of the MS “Exploring the Anti-Osteoporotic Potential of Daucosterol: Impact on Osteoclast and Osteoblast Activities”, the authors have almost satisfactorily answered my points of criticism. I have only two minor comments.
Fig 4B. Write: Area taken up/total area (%) or Pit area/total area (%)
Line 564: It should read “patellar surface”
Author Response
Responses to Reviewer:
We would like to thank the reviewer for your positive and very helpful comments to improve our manuscript. In response to reviewer, we have addressed the comments as follow:
In the revised version of the MS “Exploring the Anti-Osteoporotic Potential of Daucosterol: Impact on Osteoclast and Osteoblast Activities”, the authors have almost satisfactorily answered my points of criticism. I have only two minor comments.
- We believe that your comments have greatly improved the quality of the manuscript. We thank you and have provided responses to your comments below.
- Fig 4B. Write: Area taken up/total area (%) or Pit area/total area (%)
We modified the contents of the Y axis of the figure 4B.
- Line 564: It should read “patellar surface”
- We corrected typos in the manuscript.